# Characterization of Communes with Quality Accredited Primary Healthcare Centers in Chile

**DOI:** 10.3390/ijerph19159189

**Published:** 2022-07-27

**Authors:** Juan Coss-Mandiola, Jairo Vanegas-López, Alejandra Rojas, Raúl Carrasco, Pablo Dubo, Maggie Campillay-Campillay

**Affiliations:** 1Facultad de Ciencias Médicas, Escuela de Obstetricia y Puericultura, Universidad de Santiago de Chile (USACH), Santiago 8320096, Chile; jairo.vanegas.l@usach.cl (J.V.-L.); alejandra.rojas.r@usach.cl (A.R.); 2Facultad de Ingeniería y Negocios, Univerdidad de Las Américas, Santiago 3981000, Chile; rcarrasco@udla.cl; 3Departamento de Enfermería, Facultad de Ciencias de la Salud, Universidad de Atacama, Copiapó 7500015, Chile; pablo.dubo@uda.cl (P.D.); maggie.campillay@uda.cl (M.C.-C.)

**Keywords:** health facility accreditation, health centers, primary health care, quality of health care, local government

## Abstract

The accreditation process of primary healthcare centers in Chile has not had the same progress as in hospitals, which show high levels of compliance. The purpose of this research is to characterize the communes that have accredited family healthcare centers (CESFAMs) through socio-economic, municipal management, clinical management, and population variables by performing a principal components analysis (PCA) with biplot analysis and a grouping of communes through a hierarchical analysis. The biplot analysis and hierarchical analysis yielded the formation of three large groups of communes with accredited CESFAMs, characterized mainly by population size, number of people registered in the municipal health system, socioeconomic indicators, and financial management and clinical management variables. It was found that the communes that have accredited CESFAMs are characterized by dissimilar behavior in relation to the variables analyzed. Through the model used, it was possible to establish at least three groups of communes according to their behavior against these variables. Of these, the variables of a municipal financial nature were not decisive in achieving the accreditation of the CESFAMs of these communes. Therefore, it is possible that there are other variables or factors that could be facilitating the achievement of accreditation processes.

## 1. Introduction

According to the World Health Organization (WHO), primary care is where most of the benefits are delivered to the population, and the functions addressed are essential to public health. In this context, the quality of care is essential to ensure safe health outcomes and respond to the needs and expectations of the population [1].

The Chilean health system is mixed. it is made up of public and private funds and providers [2]. The care model is supported by the primary care network made up of four types of establishments; they are: 593 family health centers (CESFAMs) [3], 218 community family health centers (CECOSF), 1166 rural posts, and 104 community hospitals, which, regardless of their source of financing, implement the guidelines provided by the Ministry of Health. It should be noted that, of the total CESFAMs, 95% of them are managed by municipalities (local government), 3% are managed by Health Services, who act as managers of territorial health networks (central government), and the remaining 2% are managed by other non-profit corporations [4].

Avedis Donabedian [5], in 1990, described the seven pillars that support quality of care: efficacy, effectiveness, efficiency, optimization, acceptability, legitimacy, and fairness. These criteria reflect the “values and goals in force in the system of health care and in the broader society of which they are a part” [6]. Although quality measurement models are global in scope, they differ between countries depending on the effectiveness of established clinical management processes [7,8].

The literature on health management reveals that promoting quality is key in health centers of primary health care (PHC) and requires tools of evaluation and clearly defined accreditation processes, which allow the establishment of levels of constant improvement in the organization [8,9,10]. Countries such as the United States, Australia, Canada, the United Kingdom, and New Zealand have accreditation models of well-developed APS that have been built using as reference the models of quality that were initially implemented in tertiary care centers [8]. In this sense, most of the reviewed literature deals with accreditation models and systems of quality that guide towards tertiary care practices [10]. Investigating the accreditation of PHC health centers contributes to a greater understanding of this process and allows critical reflection on the standards of clinical practice implemented [11]. At this point, regardless of the model followed, the essential goal is to achieve benefits for patients [12].

Syengo and Suchman [13], in their study on innovative experiences in the quality of healthcare in Kenya, concluded that quality processes can contribute to manage the costs associated with care services.

According to Domínguez et al. [14], in Chile, even though the national policy of health has been moving from a biomedical perspective focused on tertiary care towards a biopsychosocial perspective focused on primary care, the programs that are carried out in the family health centers (CESFAMs) have not been consistent with the comprehensive family and community health model (MAIS) implemented in 2005. This aspect is relevant in the accreditation process, since it must respond to the criteria and principles declared in the MAIS.

In the national health legislation, Law No. 19937 established quality assurance in health care, which in its article 11 requires compliance with minimum standards for operation of health providers, whether public or private. Under this vision, the concept of accreditation corresponds to the “periodic evaluation process with respect to compliance with these minimum standards, according to the type of establishment and the complexity of benefits” [15]. Continuing with the idea, Decree No. 3 of the Ministry of Health also establishes that institutional providers of primary care of low complexity must undertake to submit to the accreditation procedure on a voluntary basis [16], according to the regulation that defines the standards and norms of the process [17].

In practice, according the regulations of the accreditation system for institutional health providers, in Article 5, states that “the standards will cover all matters that affect the security of the respective health benefits, such as sanitary conditions, safety requirements for facilities and equipment, maintenance and their calibration.“ In addition, they must refer to the techniques and technologies applied to benefits, personnel necessary to carry them out, their job qualification and coverage, compliance with care protocols, and other aspects related to matter that are necessary for the purpose of safeguarding the security of the users. Article 6, describes that “general standards will be established for the different types of establishments and specific ones for certain services or groups of benefits” [18].

In the case of the CESFAMs for the purposes of the accreditation regulations, they are considered institutional providers of open (ambulatory) care, and the specific technical manual breaks down their demands and requirements into areas, components, characteristics, and verifiers. The areas to be evaluated in the accreditation process are the following [19]: respect for the dignity of the patient, quality management, clinical management, access, opportunity and continuity of care, human resource competencies, records, equipment safety, facility security, supporting services.

Regarding the achievement of CESFAM accreditation, according to data from the Superintendency of Health, as of December 2019, only 33 CESFAMs had been accredited, a figure that represents 5.6% of the total number of providers of this type (ambulatory and low-complexity care) [20]. This figure exposes very low levels of accreditation in this type of providers, given that in Chile there are 2704 public health establishments belonging to the National System of Services of health, of which, 95% correspond to primary care establishments and of them, 593 are CESFAMs [21] (See Table 1).

The purpose of this study is the characterization of the establishment facilities of primary health care (CESFAMs) accredited in quality in Chile, in relationship to socio-economic and municipal management variables from the communes to which these accredited establishments belong. It is intended to contribute with new background, which could be influencing the good results obtained by the accredited centers.

Specifically, it is expected:Classify accredited CESFAMs according to socioeconomic, demographic, and municipal management variables;Identify interdependence between the communes that have accredited CESFAMs, according to the variables under study;Analyze the groups of communes that have accredited CESFAMs by applying hierarchical methods.

## 2. Materials and Methods

A cross-sectional study was proposed using data from municipalities that have CESFAMs accredited in quality as of 2019. In total, 17 communes were considered, which have 33 accredited CESFAMs [22]. The variables considered were socio-economic, demographic, and management variables (clinical management, human resources (HR) management, and financial management) of the municipalities (Table 2). The source of these data can be found in the National Municipal Information System with data to 2019 [21].
(1)$${dist}_{\mathrm{canberra}}=\sum_{i=1}^{n}\frac{|x_{i}-y_{i}|}{|x_{i}\left|+\right|y_{i}|}$$

The variables were selected avoiding the redundancy of those that were in perfect correlation or with high collinearity. Subsequently, an analysis was carried out with the PCA statistical method, which allows the simplification through dimension reduction [25] of the complexity of sample spaces with many dimensions or variables. This model seeks to maximize the variance extracted by the new variables (components). The higher the variance extracted, the more information is kept in the new variable, therefore making the solution better. Then, a biplot analysis based on PCA was generated [26,27] to identify and analyze how the dimensions and observations converge. After the extraction of the components, hierarchical cluster analysis was applied, the second technique applied in this study with original standardized variables, to compare the results obtained and observe whether different groups were formed (See Figure 1). Given that the metric distances quantitatively formalize the similarity and dissimilarity between cases, limitations may be observed, since they are affected by the unit of measure implemented with the use of the applied Canberra distance (Equation (Equation 1)). The variables measured in large magnitudes cancel the effects of variables with a small amplitude range. Therefore, the commonly used transformation is the standardization of the data.

The software R version 4.2.1 was used in this research work as a statistical analysis tool [28].

## 3. Results

Principal component analysis (PCA) shows dimensional reduction obtained [29]. The first two dimensions represent 63.8% of the total variability of the system (See Figure 2).

According to Figure 3a, Dimension 1 represents 41.7% of the variability of the model, and is composed mainly of municipal financial management variables, clinical management, HR management, and demographics: operating expenses of the health sector (ISAL021), total number of contracted physicians as of December 31 (MTFCM), population census 2017 (POPULATION), expenditure on personnel in the health sector (ISAL019), registered population validated in municipal health services—FONASA (HPISM), number of staff health sector (MPSP), morbidity consultations made to adolescents between 10–19 (MTCM1019), and expenditure on personnel to health sector by contract (ISAL031). In Figure 3b, Dimension 2 represents a 22.1%, and is made up of socioeconomic-type variables, multidimensional poverty, clinical management, and financial management: annual expenditure of the health area per registered inhabitant validated (ISAL23), municipal contribution to the health sector per registered person validated (APMUNI), percentage of people in a situation of multidimensional poverty (POBRMULT), municipal transfers to health on municipal own income (ISAL016), percentage of people in a situation of income poverty (POBRINCOME), contribution of the MINSAL (per capita) with respect to the total income of the health sector (ISAL012), and morbidity consultations made to adolescents between 10–19 (MTCM1019) (see Table 2).

In Figure 3a,b, it can be seen that PORMPV as a clinic management variable does not contribute information to the construction of dimensions 1 and 2. On the other hand, we observe that the variable MTCM1019 of clinical management is the only variable that contributes to the formation of dimensions 1 and 2.

Regarding the grouping of communes, the set that concentrates the largest number of these, with six of them, representing 35%, includes Constitución, San Felipe, Chillán Viejo, Tomé, Coronel, and Penco. These communes are mainly influenced by poverty rates and population size (See Figure 4).

A second grouping of communes is found at the crossroads of dimensions 1 and 2. In this group, we find the communes of Talcahuano, Macul, Valdivia, and Chillán, representing 23.5% of the total number of communes studied. These communes are grouped in the center of Figure 4, since they are around averages of the variables and dimensions represented.

In Figure 4, we also observe that the communes of Rancagua and Peñalolén are mainly influenced by dimension 1, which represents communes with a larger population and a larger registered population validated in the municipal health services. On the other hand, the communes of Ñuñoa and Las Condes are influenced by low rates of poverty, high concentration of population, and a low proportion of population registered validated in its municipal health services, with a high level of municipal contribution and high annual expenditure per registered inhabitant. Finally, Pirque has a low population concentration, low poverty rates, and a high level of population registered and validated in municipality health systems, with a high level of rural population.

Figure 5 shows a heat map between the communes and the variables representing the distance matrix in its different shades for communes and municipal variables. A total of three large clusters are formed:1.Penco, San Felipe, Constitución, Tomé, Chillan Viejo and Pirque: this agrees with previous analysis as the communes with the lowest population concentration;2.Coronel, Macul, Valdivia, Chillan, Talcahuano and Concepción: these are communes with values close to the average of the variables;3.Ñuñoa, Las Condes, Lo Prado, Peñalolen y Rancagua: these are grouped together because they are communes with high concentration of population, while Las Condes and Ñuñoa differ as communes with low levels of poverty, a low proportion of registered population validated in municipal health systems, and high levels of financing of health systems.

## 4. Discussion

The results obtained in this study provide information on the characterization of communes that have accredited CESFAMs and their link with some variables of clinical, financial, HR, and socio-economic management, among others. The foregoing is mainly translated into the identification of certain groupings of communes formed around the variables under study. Although it is not possible to establish a causal relationship between the variables studied and the accreditation of the CESFAMs in the communes studied, the research shows the relevance of socioeconomic and municipal management variables for the accreditation of said PHC establishments in these communes.

The aforementioned is relevant considering that the World Health Organization (WHO) has highlighted primary health care (PHC) as a cornerstone to comply with the 2030 sustainable development agenda. Therefore, strengthening systems to improve quality at the first level of care is one of the essential axes to generate gradual changes and strengthen the health system of countries [30].

The first contact of a sick person or a person who requires a health service is with primary health care (PHC), through which the patient enters the Chilean public health system. Therefore, family health centers (CESFAM) are at the basis of operational integrity, continuity, and resolution, creating a continuity of care between PHC, hospitals, and other health service providers [31].

The health reform carried out by Chile in 2005, which promoted the comprehensive health model with a community approach, and the persistence of hospital-centrism, characteristic of the biomedical model, has favored an increase in inefficiency and great inequality in the system [32,33]. In effect, this has caused primary care programs to occupy a subordinate role with respect to the secondary and tertiary levels [34]. However, despite this, there is perceived inequality in health determined by socioeconomic and geographic gradients, among other possible factors.

Studies at the international level acknowledge that inequalities in health and access to medical care vary along social gradients. However, the specific allocation of health resources can reduce the range of disparities [35]. Regarding this, this research clearly shows the strong influence of two financial variables, such as the annual expenditure per inhabitant and the municipal contribution to the health sector, which strongly influence the communes of Ñuñoa and Las Condes, which are also communes with a higher socioeconomic level. The foregoing allows us to understand the influence exerted by these variables on the formation of groups of communes with different socioeconomic levels and with accredited CESFAMs.

Poor health can be indirectly measured by avoidable hospitalization rates, morbidity rates, and mortality [36]. In contrast to the previously proposed model, this study shows that the variables related to social determinants allow the grouping of communes with similar characteristics, and even though some have high rates of poverty and low municipal income, these communes achieved the accreditation of their CESFAMs. This could be explained by the incorporation of a culture of quality in the officials that make up the health centers, thereby contributing to the success of the process and to better care for users, as reported by some studies [37].

Accreditation can have a favorable impact on health teams in primary care, as described in a study carried out in the Netherlands, where the application of an accreditation internship program had positive effects on the team climate, causing a greater sense of responsibility for quality of care among all team members [38]. This could be relevant from the point of view of human resources management, since, in this study, we observed that, of the seven HR variables included, five of them contribute strongly to the grouping of accredited communes. Consistent with this, a study carried out with relevant stakeholders in the accreditation process of family health centers in Concepción, Chile concluded that accreditation is an important factor for services to be provided with quality, thus improving the safety of users and their expectations, with the most relevant factors being the commitment of managers and more human and structural resources [37].

On the other hand, in Chile, quality accreditation systems at the tertiary level of health care obtained levels of compliance above 80% in the period between 2018 and 2020 [39], contrasting with the low levels of accreditation obtained in primary health care (APS). The percentages of accreditation are concentrated especially in three of the sixteen regions of the country: the Bío-Bío region, the Metropolitan region–national capital, and the Ñuble region, which have high population density except for the last one and are geographically located in the center-south of the country. Studies in this regard have described that the larger the APS health center, the better the results obtained in the quality accreditation processes. At this point, there could be a relationship with the higher level of complexity that a large center reaches, associated with the quantity of available resources, human resources, infrastructure, and processes that facilitate its progress [40]. However, various studies reveal that these are centers with smaller communities that are disadvantaged in the system, which generally have a greater burden of poor health, generating demand for reactive care that is more expensive instead of preventive care, and thereby producing disparity in health care [41].

The data analyzed in our study allowed us to observe that the communes that have accredited CESFAMs are highly populated, with more than 100,000 inhabitants, and are preferably located in urban areas. Comparatively, the study carried out by García Huidobro et al. [42] in Chile evaluated the progress of MAIS in PHC centers, observing the same trend. That is, urban centers and communes with a larger registered populations (beneficiaries) present better results in the MAIS, something that could be attributable to certain communes being able to allocate more resources to primary health.

This last aspect can be complex to explain, considering that primary health centers are financed with the per capita system. This financing model establishes an amount assigned for each validated registered beneficiary, which, analyzed by experts, requires adjustment variables, especially due to the socioeconomic situation of the users and rurality [43,44]. This form of financing is complemented by municipal contributions, which in practice means that richer communities with fewer health demands have more resources to deliver the services of the basic basket of the family health plan [4,44,45]. This aspect sheds light on the fact that the priorities of CESFAMs are benefits related to the delivery of activities, while in aspects related to quality management or control, this accounts for a “fight for an efficient and equitable allocation of public resources” [9].

Although our study does not allow us to establish a relational line between the variables studied, it can show the strong influence that the group of financial management variables, such as the annual expenditure of the health area per inhabitant registered, has on grouping these communes. On this, a study in the field of maternity care in PHC in Indonesia found that a facilitating factor for accreditation turned out to be budget availability, which was managed by the team leader [37,46].

Despite the strong influence that financial variables have on achieving accreditation, a study carried out in Colombia and published in 2012 indicates that, although accreditation requires economic investment, it is not exclusive to institutions that have resources available to implement it [47]. This study adds that the accreditation process is linked to generating a culture change in organizations that provide health services and improving the behavior of the human group that works in it, among other actions [47].

According to what was analyzed by Alvial et al. [48], the lack of unification in the administration and management of primary care services has generated limited access of patients to clinical services at that level of care, consequently causing an overuse of emergency services in the tertiary sector, saturating the system. This means that the principle of MAIS has been displaced, since primary care centers are not enough to carry out health promotion and prevention activities, but rather, have had to provide clinical care of an assistance type, with even this being insufficient within the Chilean Health System. Regarding this, in the analyzed data, a dissimilar behavior is observed between the variables of the clinical management type. On the one hand, we find that the variable related to PAP coverage in women aged 25 to 64, which is a preventive type benefit, contributes little information to the model. On the contrary, the variable related to morbidity consultations in adolescents aged 10 to 19 years, which is a clinical care provision, provides more information to explain the model. Garcia Huidobro et al. [42] mention that even though the MAIS in Chile has achieved a high level of implementation of the axes related to technologies, health promotion, and community participation, on the contrary, the axes of family focus and quality have reached less development, as aspects that have not been studied at the national level.

In relation to the heterogeneity of CESFAMs in Chile regarding their level of complexity [49], geographic location, structural characteristics, number of beneficiaries, funding levels, and other administrative variables, Díaz and Galán [9] consider that these “contextual and structural differences in health centers would not make the achievement of quality, goals, good practices, accreditation and excellent care comparable to transversal levels”. This places the management of people at the center of the discussion, since, in such varied environments, the management skills of the directors, the variables, the social skills of the staff, and the motivation and leadership of the interdisciplinary teams of primary care are highly relevant both in the planning of improvements, the result of an accreditation process, and in the improvements that are essential to develop a culture of quality management [9,10,50,51].

According to Díaz Herrera and Galán Torres [9], regardless of the structural conditions enabling quality, such as necessary spaces and technological implementation, as well as adequate officials prepared to certify and accredit, it is essential that the management bodies comply with management requirements and possess soft skills sufficient to become leaders that can carry out the necessary changes in a health center. Based on the latter, this study reveals the importance of the variable expenditure on training personnel in the health area, through which it is possible to favor the improvement of the competencies of the managerial personnel to better support the accreditation processes.

Regarding the above, our study reveals that one of the large groupings of communes, which brings together 6 of 17, corresponding to group 1 of the hierarchical analysis, is influenced by the variable percentage of expenditure on health personnel over total expenditure of health, considered as a variable of HR Management. This allows us to suggest that a high percentage of the communes that have accredited CESFAMs can be characterized by the importance they assign to their HR. This same condition is ratified by the superintendent of Health, which points out that managerial commitment, the empowerment of leaders, and quality training are must involve all the staff of the institution [52].

Similarly, a study carried out in the PHC of Indonesia concluded that the factors that favor accreditation at the organizational level include the integration of accreditation in the structure of the organization, the culture, and the activities, ensuring financial resources and investing in human resources [46].

Although the study cannot account for relational aspects between these variables, it at least provides some clues regarding factors that may be critical when a CESFAM undertakes the challenge of an accreditation process. We consider that these critical success factors in quality environments are aspects that favor processes and generate competitive advantages. For this reason, we agree with various authors that it is necessary to specify these key areas or processes, since this is precisely what can positively affect the results of accreditation processes. However, these critical factors cannot be confused with objectives or goals, since they are in themselves variable and, when identified and managed appropriately, make it possible to achieve comprehensive results and the fulfillment of corporate objectives, which, in this case, are is related to the achievement of accreditation [53].

Finally, regarding the external validity of the study, it is easily generalized and applicable, even as the communes achieve accreditation of their CESFAMs.

## 5. Limitations

The main limitation of the study was not being able to have complete data on everything with respect to the variables, indicators, or characteristics of the accreditation that are involved in the process. We understand that this occurs due to confidentiality of the information on the accreditation process, of which only the percentage achieved by each institutional provider that manages to be accredited is available.

Not having the results of the evaluation according to the specific indicators of accreditation does not allow us to carry out analysis of association with the variables of this study and the accreditation. Having it would be relevant for the formulation and application of public policies that could strengthen certain areas of municipal health management that are related to the accreditation of quality in health.

## 6. Conclusions

Chile has enacted a health reform since 2005, promoting a comprehensive health model with a community focus and PHC. In practice, however, the emphasis placed on primary care has not favored the management processes at this level of care, including in the quality accreditation of its health establishments. Proof of this is that until 2019, only a small number of communes had accredited CESFAMs.

This delay in the PHC accreditation process has direct consequences for the beneficiaries of the system considering its contribution to improving the quality of health care; therefore, its postponement could affect the quality of the services that are offered to the population.

From the analysis of the data, we can infer that the communes that have accredited CESFAMs are characterized by a dissimilar behavior against the variables studied. This does not allow us to affirm that one or another variable directly influences the achievement of accreditation. However, it is worth noting that a group of communes has a low proportion of registered population validated in their municipal health system, have high levels of income and expenses, and have low levels of poverty, which could have influenced their accreditation process. The same occurs for other communes, in where the low population concentration could be relevant.

From the foregoing, it could be suggested that the variables of a financial nature were not decisive in achieving the accreditation of the CESFAMs of these communes. However, it is possible that there are other factors that could be facilitating the achievement of accreditation processes.

Finally, some implications that could be derived from the study could contribute to improving public policies related to quality management and accreditation, especially for communes that do not have accredited CESFAMs, since knowing adequately how communes with establishments that did manage to comply with said process behave can lead to improvements in municipal management policies that favor the accreditation process in PHC establishments. Another of the political implications is the need to overcome the limitation in open access to the details of the information obtained in the accreditation processes, as well as the economic limitations in which the establishments find themselves. These problems have resulted in the fact that mainly those CESFAMs with a strong teamwork component are the ones that have successfully obtained accreditation. Therefore, it is suggested to include qualitative analyses that make it possible to determine certain facilitating factors or obstacles to the quality accreditation process in PHC establishments.

## Figures and Tables

**Figure 1 ijerph-19-09189-f001:**
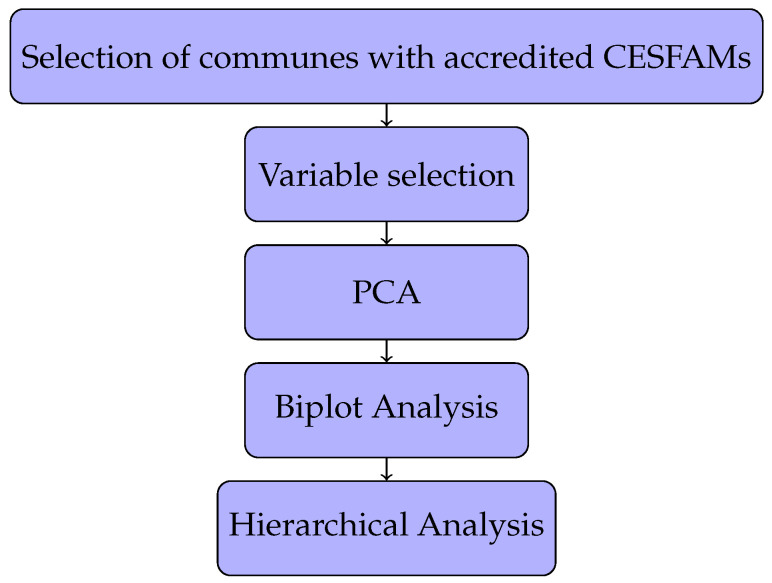
Research flow chart.

**Figure 2 ijerph-19-09189-f002:**
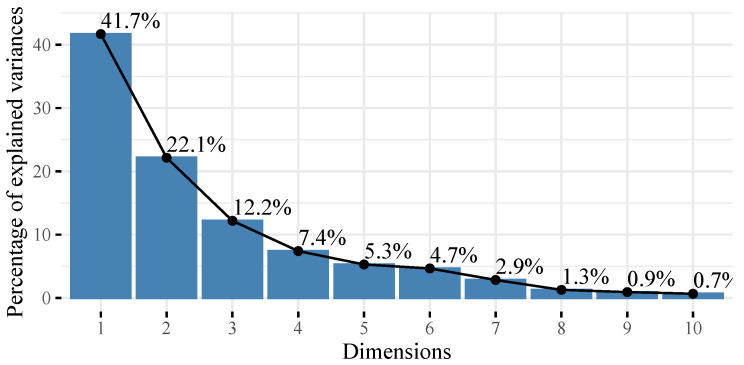
PCA plot.

**Figure 3 ijerph-19-09189-f003:**
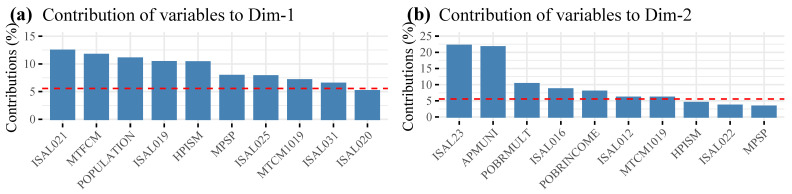
Contribution of variables to: (**a**) Dim. 1, (**b**) Dim. 2.

**Figure 4 ijerph-19-09189-f004:**
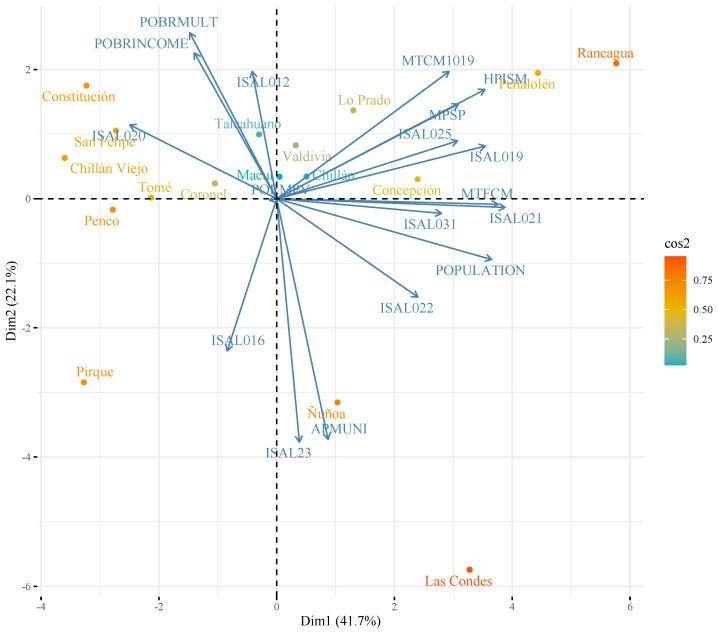
Biplot.

**Figure 5 ijerph-19-09189-f005:**
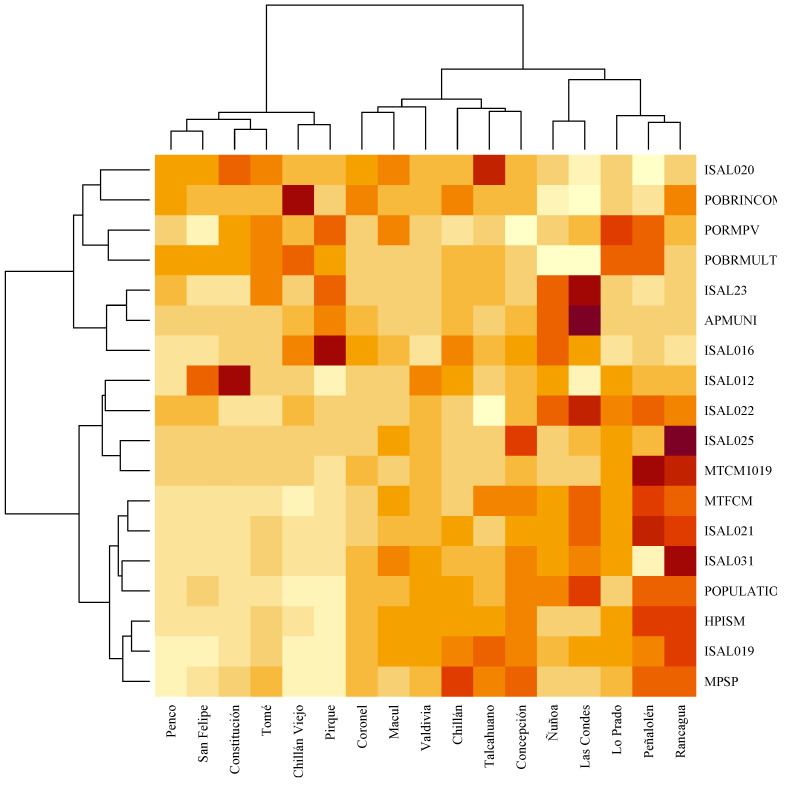
Hierarchical analysis with heat map for communes and variables; method—ward.D; distance—Canberra.

**Table 1 ijerph-19-09189-t001:** Summary of CESFAMs accredited in Chile as of December 2019.

Accreditation	No	Yes	Total	% No	% Yes
CESFAM ^1^	560	33	593	9444	556
Communes ^2,3^	255	17	272	9375	625
Regions	9	7	16	5625	4375

^1^ According to Department of Health Information Statistics (DEIS), July 2019, http://www.deis.cl/, (accessed on 31 May 2022). ^2^ Of a total of 346 communes, 272 have CESFAMs, according to DEIS, July 2019, https://www.supersalud.gob.cl/acreditacion/673/w3-propertyvalue-4710.html, (accessed on 31 May 2022). ^3^ Commune: unit of local government used in our territory of study.

**Table 2 ijerph-19-09189-t002:** Description of variables, taken from [21,23,24].

	Variable	Description	Unit	Class
^1^	ISAL012	Contribution of the MINSAL (per capita) with respect to the total income of the health sector	%	Financial management
	APMUNI	Municipal contribution to the health sector per registered person validated	M$	Financial management
	ISAL016	Municipal transfers to health on municipal own income	%	Financial management
	ISAL019	Expenditure on personnel in the health sector	M$	HR management
	ISAL020	Percentage of expenditure on health personnel over total expenditure	%	HR management
	ISAL021	Operating expenses of the health sector	M$	Financial management
	ISAL022	Percentage of operating expenses over total health expenditure	%	Financial management
	ISAL025	Expenditure on personal training health area	M$	HR management
	ISAL031	Expenditure on personnel to health sector by contract	M$	HR management
	ISAL23	Annual expenditure of the health area per registered inhabitant validated	M$	HR management
	PORMPV	Percentage of women between 25 and 64 years with current PAP	%	Clinical management
	MPSP	Number of staff in health sector	No.	HR management
	MTCM1019	Morbidity consultations made to adolescents between 10–19	No.	Clinical management
	MTFCM	Total number of contracted physicians as of December 31	No.	HR management
	HPISM	Registered population validated in municipal health services (FONASA)	No.	Demographic
^2^	POBRINCOME	Percentage of people in a situation of income poverty	%	Socio-economic
	POBRMULT	Percentage of people in a situation of multidimensional poverty	%	Socio-economic
^3^	POPULATION	Population census 2017	No.	Demographic

^1^ SINIM, (accessed on 31 May 2022), http://datos.sinim.gov.cl/datos_municipales.php. ^2^ Social Observatory, (accessed on 31 May 2022), http://observatorio.ministeriodesarrollosocial.gob.cl/pobreza-comunal-2017. ^3^ Census 2017, (accessed on 31 May 2022), http://www.censo2017.cl/descargue-aqui-resultados-de-comunas/.

## Data Availability

Not applicable.

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
