# Peer review of "Characterization of Communes with Quality Accredited Primary Healthcare Centers in Chile"

_ijerph, 2022, doi:10.3390/ijerph19159189_

Round 1

Reviewer 1 Report

This study developed the communes that have accredited Family Healthcare Centers (CESFAM) through socio-economic, municipal management, clinical management and population variables; by performing a principal components analysis (PCA), with biplot analysis and a grouping of communes through a hierarchical analysis. Findings represented a contribution to this area of literatures and were strengthened by this paper.

Substantial areas for clarification/consideration are noted below. The motivation of the study and policy implications need some further work. The paper would benefit from some further proofreading as there are types and grammatical errors. The authors may also make it clear in the text when describing the different trainers.

My detailed comments on the paper ( in the order they appear in the paper) are given below.

1. Please provide specific bullet pointed aims at the end of the introduction i.e., detail each research questions separately.

2. Please explain a capacity building program to examine analysis the association between characterization of communes and quality accredited primary healthcare in your manuscript.

3.The software used for all analyses should be included.

4. The authors should make the policy implications clearly about your study.

Author Response

Dear Reviewer

We send the answer to your review.

We have accepted the observations made.

Best regards. 

Reviewer 2 Report

Manuscript ID ijerph-1798538: Characterization of communes with quality accredited Primary Healthcare Centers in Chile

Int. J. Envirom. Res. Public Health

The authors present a quantitative study aimed to characterize the communes that have accredited Family Healthcare Centers (CESFAM) through socio-economic, municipal management, clinical management and population variables; by performing a principal

components analysis (PCA), with biplot analysis and a grouping of communes through a hierarchical analysis. I found this article a strong piece of evidence with just minor reviews needed. In general, there are some mistakes in written English (syntax, spelling, and grammar) that would need to be addressed and revised by a native English speaker.

Title and abstract:

Please clarify the specific definition of “communes”.

Introduction:

I agree with the authors that this study fills a gap in the literature and makes up for deficiencies in existing studies on the topic. I think it is worthy of being published, with major revisions.

The authors provide scientific background and an explanation of the rationale.

The introduction is well developed, however, a bit long. I would suggest shortening it up a bit.  

Bullet points can be removed (lines 88 to 95), and the info can be described as text.

Table 1 can be removed and its information can be simply described as text.

Restructure the introduction using the following template:

  1. Background: Introduce the area/field in which your article takes place, highlighting the status of our understanding of the problem you are tackling. Give your readers the essential technical details they need to understand the problem–nothing more. Explain the scientific background and rationale for the investigation being reported
  2. What is not known, what is the knowledge gap? - state you problem: After discussing what you know, articulate what you do not know, specifically focusing on the question that has motivated your work.
  3. How you tried to answer the issue/“Here we show…”: Very briefly summarize why you put this article together, the objective and how you are trying to answer the problem you stated.

Methods:

Strong methodology, however, the overall description of the methods section lacks details. See below a few suggestions/clarifications (based on STROBE checklist for cross-secional studies):

       How was eligibility criteria determined (rationale)? Give the eligibility criteria, and the sources and methods of selection of participants.

       For each variable of interest, give sources of data and details of methods of assessment (measurement). Describe comparability of assessment methods if there is more than one group.

       Describe any efforts to address potential sources of bias.

       Explain how missing data were addressed.

Results:

No comments.  

Discussion:

       Start the Discussion section by restating your main objective and by summarizing the main results (all in the 1st paragraph). Summarise key results with reference to study objectives.

       Remove background information that was already mentioned in the introduction. Leave essential information in the discussion that will help the audience interpret your results. 

       This section is a bit long. I’d suggest giving a cautious overall interpretation of results considering objectives, limitations, multiplicity of analyses, results from similar studies, and other relevant evidence. Remove unnecessary information.

       Discuss the generalisability (external validity) of the study results. 

Limitations:

       This section is very brief, it needs more details on the study limitations.

       Discuss limitations of the study, taking into account sources of potential bias or imprecision.

       Discuss both the direction and magnitude of any potential bias

Conclusion:

This section is clear and quite concise, but there are only a few implications for future research in terms of methods, maybe some other implications should be added.

The statement in lines 383 and 384 is not quite clear. Please rewrite.

Author Response

(The authors gave the same response as above.)

Round 2

Reviewer 1 Report

I am satisfied with the atuhor's revsied manuscript and have no comments for anuthors.